

# Unusually low genetic divergence at COI barcode locus between two species of intertidal *Thalassaphorura* (Collembola: Onychiuridae)

Xin Sun[1,2], Anne Bedos[3] and Louis Deharveng[3]

[1] Key Laboratory of Wetland Ecology and Environment, Northeast Institute of Geography and Agroecology, Chinese Academy of Sciences, Changchun, China
[2] J.F. Blumenbach Institute of Zoology and Anthropology, University of Göttingen, Göttingen, Germany
[3] Institut de Systématique, Evolution, Biodiversité, ISYEB—UMR 7205—CNRS, MNHN, UPMC, EPHE, Sorbonne Universités, Museum national d'Histoire naturelle, Paris, France

## ABSTRACT

Species classification is challenging when taxa display limited morphological differences. In this paper, we combined morphology and DNA barcode data to investigate the complicated taxonomy of two Onychiurid Collembolan species. *Thalassaphorura thalassophila* and *Thalassaphorura debilis* are among the most common arthropod species in intertidal ecosystems and are often considered to be synonymous. Based on morphological and barcode analyses of fresh material collected in their type localities, we redescribed and compared the two species. However, their morphological distinctiveness was supported by a molecular divergence much smaller than previously reported at the interspecific level among Collembola. This divergence was even smaller than inter-population divergences recognized in the related edaphic species *T. zschokkei*, as well as those known between MOTUs within many Collembolan species. Our results may indicate a link between low genetic interspecific divergence and intertidal habitat, as the only biological peculiarity of the two species of interest compared to other Collembolan species analyzed to date is their strict intertidal life.

## INTRODUCTION

The intertidal zone, a narrow littoral strip between the low and high tide marks (*Mouritsen & Poulin, 2002*; *Raffaelli & Hawkins, 2012*), is a critical interface between terrestrial and aquatic ecosystems (*Raffaelli & Hawkins, 2012*). It is characterized by daily cycles of submersion and exposure due to tidal movements. Environmental conditions in this ecosystem are therefore very predictable but extremely variable within a day. Many groups of marine origin, as well as some of terrestrial origin, include organisms that are well adapted to these harsh environmental conditions.

Corresponding author
Xin Sun, sunxin@iga.ac.cn

Springtails (Collembola) are the most abundant and often the most diversified hexapods in the intertidal environment (*Deharveng, 2004*; *Joosse, 1976*), where they are often found in very large numbers. This has been shown for *Anurida maritima* (*Laboulbène, 1865*) and several species of *Thalassaphorura Bagnall, 1949* (*Christiansen & Bellinger, 1988*; *Willem, 1925*; *Witteveen & Joosse, 1988*). The genus *Thalassaphorura* is diverse and widely distributed. The taxonomic history of its intertidal species, traced in detail in *Bellinger et al. (2015)*, is complex. *Bagnall (1949)* described the genus with *Onychiurus thalassophilus Bagnall, 1937*, as the type species. A few species were subsequently described in or assigned to *Thalassaphorura* (*Fjellberg, 1998*; *Pomorski, 1998*), and various combinations and synonyms have been proposed (*Bellinger et al., 2015*). More than 60 species have been assigned to the genus until now (*Bellinger, Christiansen & Janssens, 1996–2018*), but the taxonomic status of several species, including the intertidal species of interest here, remains uncertain (*Stach, 1954*; *Kaprus & Paśnik, 2017*; *Sun, Bedos & Deharveng, 2017*). Currently, 57 valid species are recognized in the genus (*Kaprus & Paśnik, 2017*; *Sun, Bedos & Deharveng, 2017*), nine of which are halobionts or restricted to the intertidal zone (*Arbea, 2017*). Two of these intertidal species, namely, *Thalassaphorura debilis* and *Thalassaphorura thalassophila*, are widespread in the northern hemisphere. The intertidal ecology of these two species is well known (*Moniez, 1890*; *Willem, 1925*) compared to that of other species of the genus. Despite their unique habitat, the morphology of these species is similar to that of the non-intertidal species in the genus (*Sun, Chen & Deharveng, 2010*), which live in litter and soil.

Due to different placements and synonymies, the taxonomic status of the two species has been confused for a long time. *T. debilis* was described as *Lipura debilis Moniez, 1890* and *T. thalassophila* as *Onychiurus thalassophilus* in 1937. The latter was collected from intertidal habitats in Scotland and was described as a species of the "*debilis*" group, differing from others by its vestigial unguiculus (*Bagnall, 1937*). Afterwards, it was assigned as a type species of the genus *Thalassaphorura* by *Bagnall (1949)*. The generic assignation of the species was subsequently much debated. It was placed in different genera, such as *Onychiurus* Gervais, 1841 by *Stach (1954)*, *Spelaphorura* Bagnall, 1948 by *Salmon (1959)*, and *Protaphorura* Absolon, 1901 by *Gisin (1960)* and *Hopkin (1997)*, and then moved back to the genus *Thalassaphorura* by *Pomorski (1998)*. The old species *L. debilis Moniez, 1890* was assigned to *Onychiurus* by *Bagnall (1935)*, *Christiansen & Bellinger (1988)*, *Denis (1923)* and *Willem (1925)*, or to *Protaphorura* by *Hopkin (1997)* and *Jordana et al. (1997)*. *Fjellberg (1998)* synonymized the two species after studying the type specimens of *T. thalassophila* and assuming that *T. debilis* is a morphologically variable species. However, re-examination of the type material and detailed studies of fresh specimens from type localities revealed consistent differences among the two species (*Sun, Chen & Deharveng, 2010*).

The confusing taxonomy is due to insufficient detail in the earliest descriptions of the species, unjustified synonymies, the low number of distinguishing taxonomic characters and the lack of information on intraspecific variability within the species. The characters used in the taxonomy of *Thalassaphorura* are as follows: the number of

pseudocelli on the head, body and legs; the number of papillae of sensory organ of antennal III segment; the relative length of unguiculus; the length of anal spines; the number of chaetae in distal whorl of tibiotarsi; and the morphology and number of S-chaetae on the head and body (*Sun, Bedos & Deharveng, 2017*). Several of these characters are known to exhibit intra-specific polymorphism. This taxonomic uncertainty hampers meaningful studies on intertidal communities of the western Palearctic seashores, where both species are among the dominant arthropods.

In an attempt to clarify the taxonomic status of these species, we combine detailed morphological and barcode analyses of the type populations of *T. debilis* and *T. thalassophila*. In the Collembola, DNA barcoding has been used to complement morphological characters to allow species characterization in several genera, including *Deutonura* (*Porco, Bedos & Deharveng, 2010*), *Heteromurus* (*Lukić et al., 2015*), *Homidia* (*Pan, 2015*), *Lepidobrya* (*Zhang, Greenslade & Stevens, 2017*), *Protaphorura* (*Sun et al., 2017*), and *Tomocerus* (*Zhang et al., 2014*; *Yu, Ding & Ma, 2017*). DNA-based approaches are regarded as powerful tools for species delimitation, especially in groups of closely related species with uncertain taxonomic status (*Hebert et al., 2003*). Although various molecular markers have been employed at the species level, a 658-base fragment of the mitochondrial gene cytochrome c oxidase I (COI), which is widely used for barcoding animals (*Hajibabaei et al., 2007*), has been effective in most zoological groups, including birds (*Hebert et al., 2004*), fish (*Ward et al., 2005*), cowries (*Meyer & Paulay, 2005*), spiders (*Barrett & Hebert, 2005*), and Lepidoptera (*Hajibabaei et al., 2006*).

Large divergences (>5%) in DNA barcode sequences provide strong support for the taxonomic separation of two putative species (*Hebert et al., 2003*). However, the extent of divergence between congeneric species varies among invertebrate groups (*Hebert, Ratnasingham & De Waard, 2003*). Insects usually have lower interspecific divergences than non-winged arthropods. For example, average DNA barcode distances between congeneric species range from 7 to 8% in holarctic Lepidoptera (*Hebert & Landry, 2010*; *Hausmann et al., 2011*) and 9.3% in Diptera (*Hebert, Ratnasingham & De Waard, 2003*), to 11.5% in Hymenoptera and 13.9% in North America Ephemeroptera (*Webb et al., 2012*). In contrast, Collembola shows much higher divergence in COI sequences between congeneric species (*Porco et al., 2012a*; *Yu et al., 2016*), with reported values ranging from 16.35 to 24.55% (Table 1). These values are similar to divergence levels between congeneric species of other non-winged soil invertebrates, such as Scolopendromorpha (13.7–22.2% in *Wesener et al., 2016*) or Lithobiomorpha (13.7–24.5% in *Stoev et al., 2013*). Furthermore, recent molecular studies on divergences within Collembolan species have revealed divergences almost as deep as among congeneric morphological species (*Cicconardi, Fanciulli & Emerson, 2013*; *Emerson et al., 2011*; *Frati et al., 2000*; *Katz, Giordano & Soto-Adames, 2015*; *Porco et al., 2012b*; *Soto-Adames, 2002*).

In this paper, we (i) re-describe and compare the two species *T. debilis* and *T. thalassophila* based on fresh specimens from their type localities, (ii) evaluate the congruence between DNA barcode and morphological data for the delimitation of the two species, and (iii) relate the unusually low genetic divergence with respect to clear morphological differences in the broader taxonomic and ecological context.

**Table 1 Sequence divergence at COI among Collembola for congeneric species pairs, after literature and the present work.**

| Reference | Family | Genus | Number of species | Mean divergence (%) |
|---|---|---|---|---|
| This work | Onychiuridae | *Thalassaphorura debilis & thalassophila* | 2 | 4.3 |
| *Katz, Giordano & Soto-Adames (2015)* | Entomobryidae | *Entomobrya* | 11 | 17.83 |
| *Porco et al. (2012a)* | Entomobryidae | *Heteromurus* | 2 | 23.02 |
| *Pan (2015)* | Entomobryidae | *Homidia* | 2 | 18 |
| *Porco et al. (2012a)* | Hypogastruridae | *Ceratophysella* | 4 | 22.66 |
| *Hogg & Hebert (2004)* | Isotomidae | *Folsomia* | 4 | 17 |
| *Porco et al. (2012b)** | Isotomidae | *Parisotoma* | 3 | 24.55 |
| *Porco et al. (2012a)* | Neanuridae | *Bilobella* | 2 | 23.19 |
| *Deharveng et al. (2015)* | Neanuridae | *Deutonura* | 4 | 18.95 |
| *Porco et al. (2010)* | Neanuridae | *Deutonura* | 5 | 20.25 |
| *Porco et al. (2012a)* | Neanuridae | *Deutonura* | 4 | 23.24 |
| *Sun et al. (2017)*** | Onychiuridae | *Protaphorura* | 13 | 16.35 |
| This work*** | Onychiuridae | *Thalassaphorura* | 7 | 19.4 |
| *Hogg & Hebert (2004)* | Sminthuridae | *Sminthurides* | 2 | 21 |
| *Porco et al. (2012a)* | Tomoceridae | *Tomocerus* | 3 | 19.60 |
| *Yu et al. (2016)* | Tomoceridae | *Tomocerus* | 2 | 20.4 |
| *Yu, Ding & Ma (2017)* | Tomoceridae | *Tomocerus* | 6 | 18.66 |

**Notes:**
* Recalculated, *Parisotoma notabilis* excluded.
** Recalculated, the MOTUs which could not be separated by morphological characters excluded.
*** Divergence between *T. debilis* and *T. thalassophila* excluded.

## MATERIAL AND METHODS

### Sampling

Sampling was done along the shores of Dalmeny in Scotland (type locality of *T. thalassophila*) and Pointe-aux-Oies in northwestern France (type locality of *T. debilis*) (Fig. 1). Both species were collected in the intertidal zone, where they lived in dense populations, in habitats characterized by very weak slope, rocky substrate, and abundant algae and barnacles on rocks. Specimens were picked up directly from under stones at low tide with a brush, or at the surface of the water after washing of gravels and stones in a plastic basin. Only *T. thalassophila* was present in the Dalmeny site, while the species co-occurred with *T. debilis* at Pointe-aux-Oies.

### DNA extraction and sequencing

We successfully barcoded 41 specimens, including 26 *T. debilis* and 15 *T. thalassophila*, from northwest France and Scotland, and 31 specimens belonging to five additional species (Table S1), in order to illustrate the interspecific divergence among non-marine species of the same genus. The species *T. zschokkei* was represented by three populations totaling 11 specimens, which were analyzed to evaluate between-populations of genetic

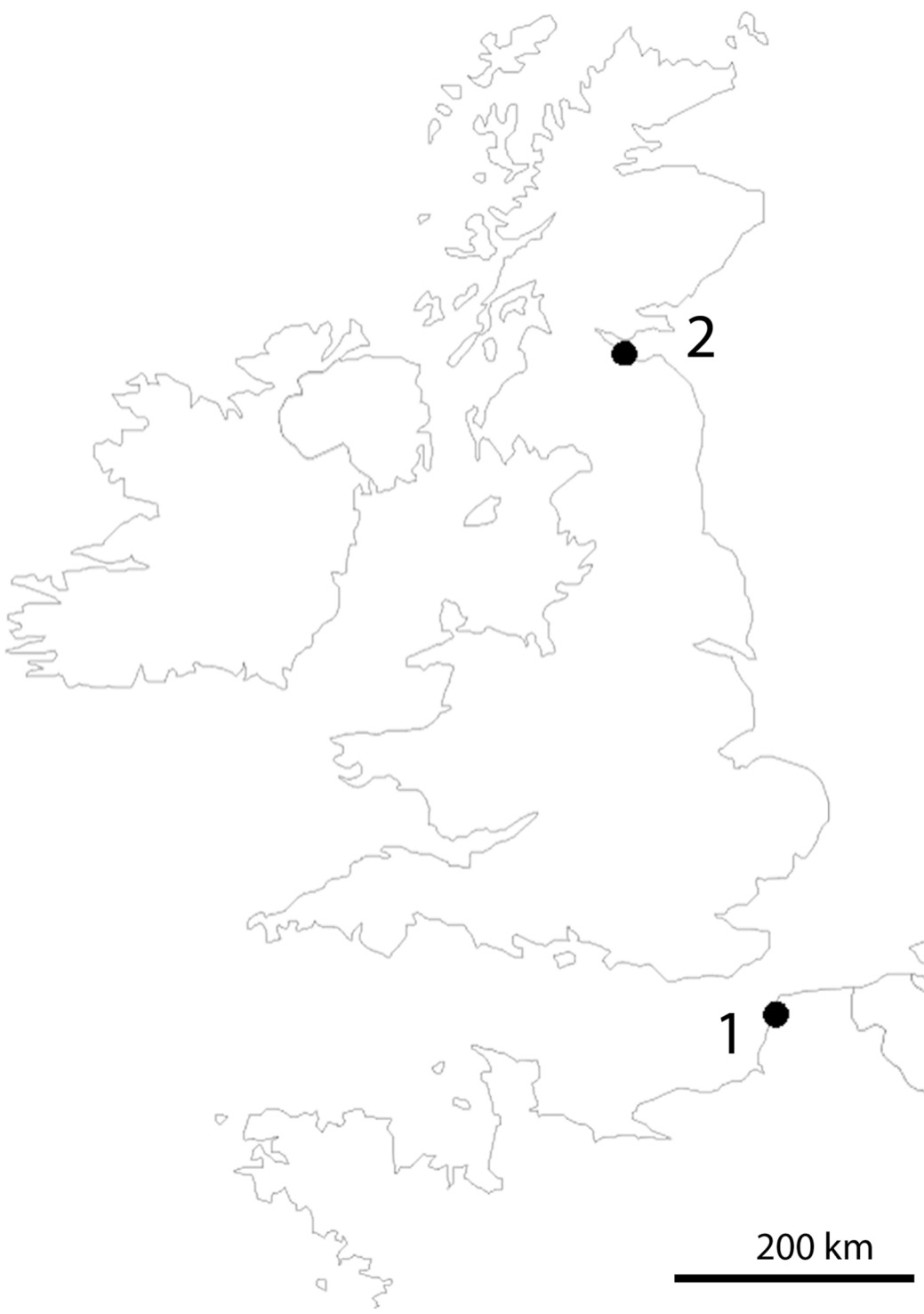

Figure 1 **Location of sampling sites.** (1) Pointe-aux-Oies in France. (2) Dalmeny in Scotland.

divergence in a non-marine species living in mountain soils and mosses. Extraction and sequencing were done at the Biodiversity Institute of Ontario, University of Guelph (ON, Canada). DNA was extracted from entire specimens in 30 mL of lysis buffer and

proteinase K incubated at 56 °C overnight. DNA extraction followed a standard automated protocol using 96-well glass fiber plates (*Ivanova, Dewaard & Hebert, 2006*). Specimens were recovered after DNA extraction for further morphological examination according to the workflow detailed in (*Porco et al., 2010*). The 5' region of COI, including 658 bp used as a standard DNA barcode, was amplified using M13 tailed primers LCO1490 and HCO2198 (*Folmer et al., 1994*). Samples that failed to generate an amplicon were subsequently amplified with a pair of internal primers combined with full-length ones, LepF1-MLepR1 and MLepF1- LepR1 (*Hajibabaei et al., 2006*). A standard PCR reaction protocol was used for amplification, and products were checked on a 2% E-gel 96 Agarose (Invitrogen, Guelph, Canada). Unpurified PCR amplicons were sequenced in both directions using M13 tailed primers (*Hajibabaei et al., 2005*), with products subsequently purified using Agencourt CleanSEQ protocol and processed using BigDye ver. 3.1 on an ABI 3730 DNA Analyzer (Applied Biosystems, Guelph, Canada). Sequences were assembled with Sequencer 4.5 (Gene Code Corporation, Ann Arbor, MI, USA) and aligned by eye using BIOEDIT ver. 7.0.5.3 (*Hall, 1999*). As we observed no indels in the COI sequences, sequence alignment was unambiguous. Sequences are publicly available on BOLD (Table S1).

## Data analyses

The K2P distances (*Kimura, 1980*) and the Neighbor-Joining tree (*Saitou & Nei, 1987*) were calculated in MEGA7 (*Kumar, Stecher & Tamura, 2016*) with 1,000 pseudo replicates and pairwise deletion and other parameters as the defaults. The frequency of K2P distances was graphed in R 3.3.2. Divergence time was estimated using *BEAST (*Heled & Drummond, 2010*). Specimens were assigned to species a priori by the results of above species delimitations. An uncorrelated lognormal relaxed clock was selected for each partition, the GTR+G+I for substitution mode and the Yule process for speciation priors. In the absence of available fossil calibrations in Collembola, the substitution rate (3.36% pairwise divergence per Mya) estimated by *Papadopoulou, Anastasiou & Vogler (2010)* was employed. An Markov Chain Monte Carlo (MCMC) chain was executed twice for 10 million generations with a sample frequency of 1,000 and the initial 5,000 generations discarded as burn-in. The effective sample size (ESS) values and convergence were checked in Tracer v1.6 (*Rambaut, Suchard & Drummond, 2014*).

## Microscopic examination

A total of 61 specimens (30 *T. debilis* and 31 *T. thalassophila*) preserved in 95% ethanol and 25 skins retrieved following DNA extraction (16 *T. debilis* and nine *T. thalassophila*) were mounted on slides in a Marc André II solution, after clearing in lactic acid. Six type specimens (the lectotype and two paralectotypes of *T. debilis* and three syntypes of *T. thalassophila*) were examined. Photos of specimens in alcohol were taken with a Jenoptik ProgRes C10+ camera mounted on a Leica MZ16. Slides were examined with a Leica DMLB microscope with DIC. A drawing was made through a *Camera lucida* and improved with Photoshop Elements 9.

## Terminology and abbreviations

Chaetotaxy of the labium, anal valves, and furca remnant is applied according to *Fjellberg (1999)*, *Yoshii (1996)* and *Weiner (1996)*, respectively. Tibiotarsal chaetotaxy is presented after *Deharveng (1983)* and is expressed as the total number of chaetae (number of chaetae in whorls A+T, B, and C, respectively). The unguiculus/unguis ratio is given according to the length of the medial line of unguiculus and the length of the inner edge of the unguis. The formulae of pseudocelli and pseudopores are presented as the number per half-tergum/sternum from head to Abd. V.

AIIIO—sensory organ of Ant. III, Abd.—abdominal segment, Ant.—antennal segment, AS—anal spine, ms—S-microchaeta, PAO—postantennal organ, pso—pseudocellus, psp—pseudopore, psx—parapseudocellus, Th.—thoracic segment, x—ventro-axial psp of Abd. IV.

## RESULTS

FAMILY ONYCHIURIDAE BÖRNER, 1913

GENUS *THALASSAPHORURA BAGNALL, 1949*

*Type species: O. thalassophilus Bagnall, 1937* (Scotland)
*Remarks on synonymies among halophilous species:*
In his reference book on Onychiuridae, *Stach (1954*: 73*)* stated that "The synonymy of the species *L. debilis Moniez, 1890* is very complicated." Although he introduced all the forms of *T. debilis* that had been validly described in his key, he expressed doubt regarding the proposed synonymies and stressed that all species "should be exactly examined." In this group with many closely related species, and in full agreement with Stach's idea, we do not accept most synonymies that have been perpetuated in the literature, as they are not supported by explicit morphological comparisons. The only exception is the synonymy *T. thalassophila = T. debilis* proposed by *Fjellberg (1998)*; however, this proposal is challenged in the present paper on combined morphological and molecular grounds. The synonymies that have to be re-assessed are the following:

*Onychiurus imminutus Bagnall, 1937* is considered a synonym of *Spelaphorura thalassophila* by *Salmon (1959*: 149*)*, based on the examination of types, but without clear justification. As the two species were collected in the same locality and are very similar, their synonymy is possible.

*Onychiurus littoralis* Dürkop, 1935 is considered a synonym of *O. debilis* by *Bagnall (1937*: 90, 145*)*, without justification.

*Onychiurus litoreus* Folsom, 1917 is considered a synonym of *O. debilis* by *Denis (1923*: 216*)*. This synonymy is challenged by *Stach (1954*: 74*)*, and the species is listed as valid by *Christiansen & Bellinger (1998*: 463*)* under the name *O. (Protaphorura) litoreus*, but without discussion of its possible synonymy.

*Aphorura neglecta* Schaeffer, 1896 is considered a synonym of *O. debilis* by *Denis (1931*: 209*)*, but not by *Stach (1954)*.

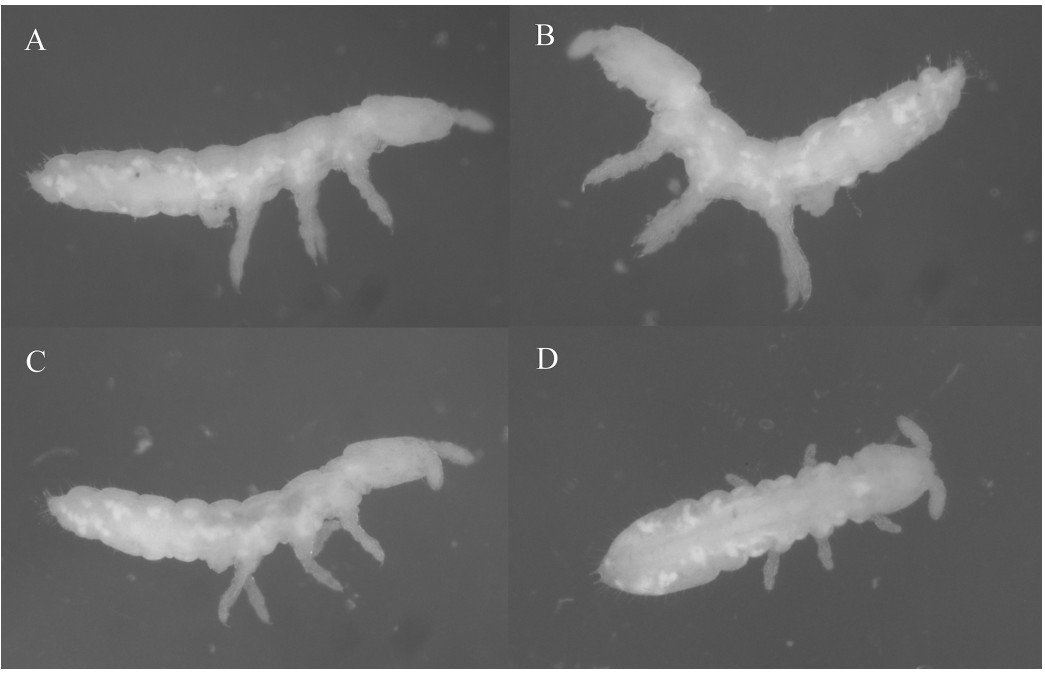

**Figure 2 Species habitus in ethanol.** (A, B) *Thalassaphorura debilis* (*Moniez, 1890*). (C, D) *Thalassaphorura thalassophila* (*Bagnall, 1937*). Photos by L. Deharveng & A. Bedos.

*THALASSAPHORURA DEBILIS* (*MONIEZ, 1890*)

(Figs. 2–4; Table 2)

*Lipura debilis Moniez, 1890*: 346

*Aphorura neglecta* Schaeffer, 1896: 112 after *Denis (1931*: 209, *syn. dub.)*

*Onychiurus litoreus* Folsom, 1917: 644 after *Denis (1931*: 209, *syn. dub.)* and *Stach (1954*: 74, *syn. dub.)*

*Onychiurus debilis* in *Denis (1923*: 216, redescription from syntypes*)*

*Onychiurus debilis* in *Willem (1925*: 279, redescription from specimens of the type locality*)*

*Onychiurus littoralis* Dürköp 1935: 133 after *Bagnall (1937*: 90, 145, *syn. dub.)*

*Onychiurus debilis* in *Stach (1954*: 73*)*

*Handschiniella debilis* in *Salmon (1964*: 162*)*

*Onychiurus (Protaphorura) debilis* in *Bolger (1986*: 193*)*

*Jailolaphorura debilis* in *Weiner (1996*: 178*)*

*Protaphorura debilis* in *Skidmore (1995*: 53*)*

*Thalassaphorura debilis* in *Fjellberg (1998*: 109*)*

*Thalassaphorura debilis* in *Sun, Chen & Deharveng (2010*: 24*)*

*Material examined:* Type material (examined). *Denis (1923)* listed eight specimens of "*Onychiurus debilis*" in Moniez's collection. Only five were retrieved in the MNHN collection. Lectotype female and two paralectotype females on slides. Label, probably

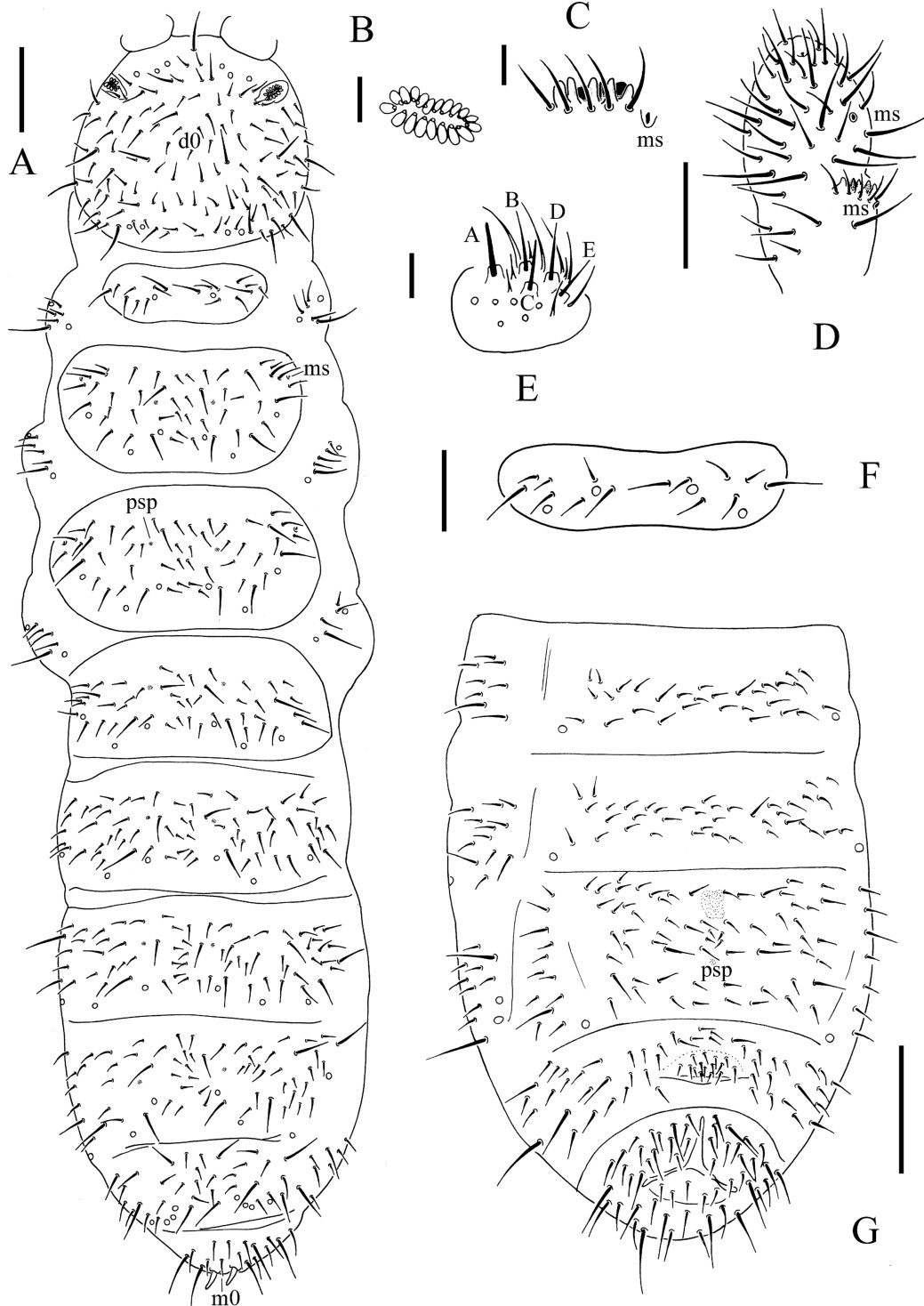

**Figure 3** *T. debilis.* (A) Habitus, pseudopores and dorsal chaetotaxy of head and body. (B) Postantennal organ. (C) Ant. III sensory organ. (D) Antennal segments III and IV. (E) Labium. (F) Th. I tergum. (G) Abdominal II–VI sterna. Scales: 0.1 mm (A, G), 0.05 mm (D, F), 0.01 mm (B, C, E).

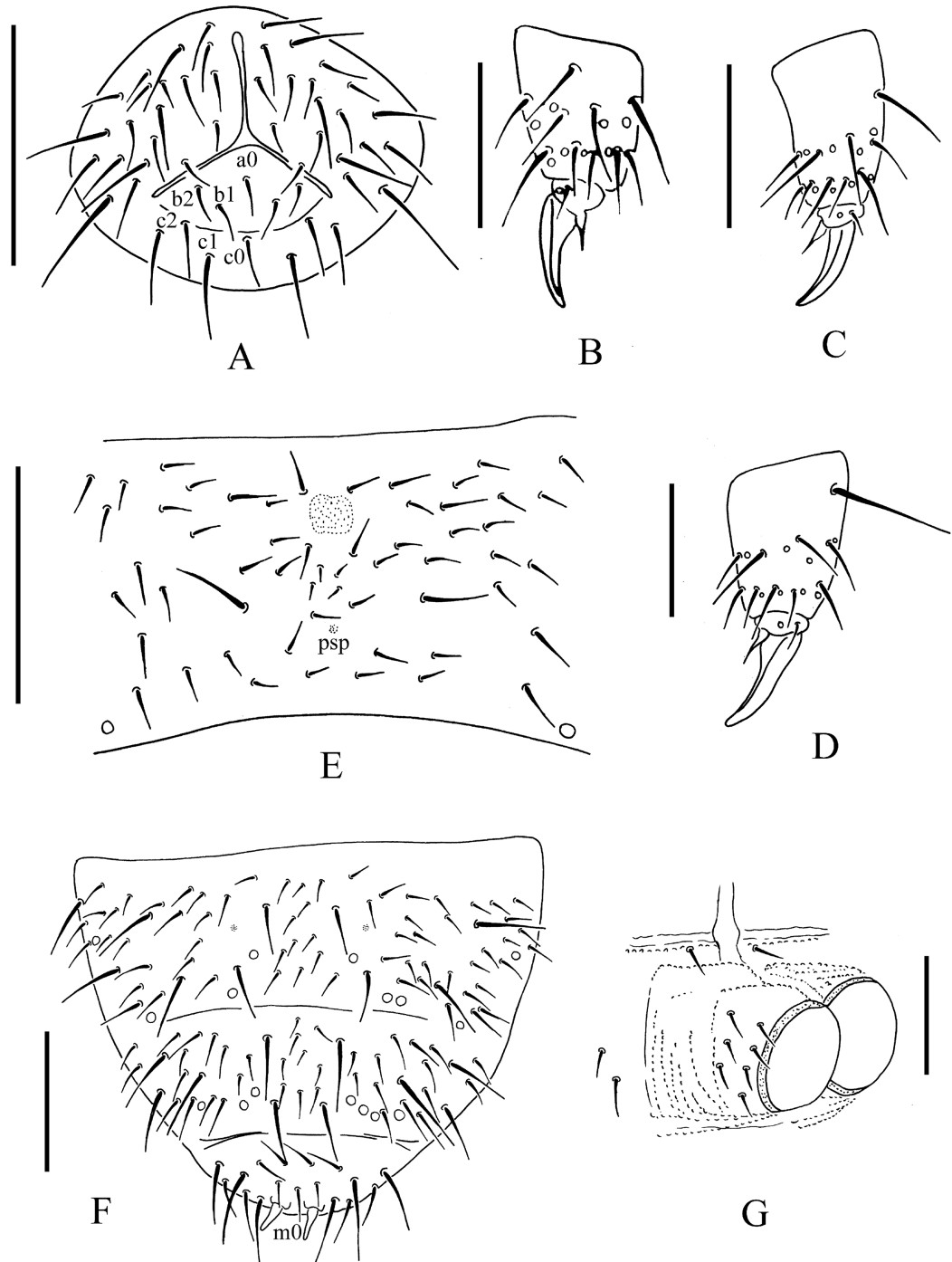

**Figure 4** *T. debilis.* (A) Anal valves. (B) Tibiotarsal chaetotaxy and claw of leg I. (C) Tibiotarsal chaetotaxy and claw of leg II. (D) Tibiotarsal chaetotaxy and claw of leg III. (E) Abd. IV sternum. (F) Abd. IV–VI terga. (G) Ventral tube. Scales: 0.1 mm (A, E, F), 0.05 mm (B, C, D, G).

re-written by Denis, as «Coll. Moniez. Pointe-aux-Oies. 2.9.89». Two paralectotypes on slides (one female, one of undetermined sex). Label, probably re-written by Denis, as «Sous les Fucus. Pointe-aux-Oies. 1.9.89».

**Table 2 Comparison of the main diagnostic characters of *T. debilis* and *T. thalassophila* from different references.**

| Source | Current conception | Current conception | Moniez (1890) (types) | Denis (1923) (types) | Willem (1925) | Jordana et al. (1997) | Fjellberg (1998) | Bagnall (1937) |
|---|---|---|---|---|---|---|---|---|
| Cited as | *T. debilis* | *T. thalassophila* | *Lipura debilis* | *Onychiurus debilis* | *Onychiurus debilis* | *Protaphorura debilis* | *Thalassaphorura debilis* | *Onychiurus thalassophilus* |
| Current name | *T. debilis* | *T. thalassophila* | *T. debilis* | *T. debilis* | *T. debilis* | *T. thalassophila* | *T. thalassophila / debilis* | *T. thalassophila* |
| Length (mm) | Female 1.4–2.1, male 1.4–1.65 | Female 1.32–1.93, male 1.20–1.66 | 1.1–1.2 | <1.5 | 0.95 | 1.5 | 1.4 | 1.5 |
| PAO | 13–21 | 16–23 | 23–28 | 19–20 | 17 | 18–19 | 15–20 | 16–20 |
| Dorsal pso formula | 32/1–233/3,3–4, 3,4–6,3–4 | 32/133/33343 | ?2/2??/???4 | 32/1$\geq$2$\geq$3/$\geq$2$\geq$2$\geq$2$\geq$23 | 32/133/33354 | 32/133/33343 | 32/133/33343 | 32/133/33343 |
| Ventral pso formula | 11/000/0111(2)0 | 11/000/00000 | ? | ? | ? | 11/000/00000 | 11/000/00000 | 11/000/00000 |
| pso on subcoxae I-III | 222 | 111 | 222 | ? | 111 | 111 | 111 | 111 |
| Axial chaetae on Abd.VI | m0 | m0 (a0) | ? | ? | ? | m0 | 1 or 2 | a0 |
| Ratio of AS/clawIII | 0.47–0.77 | 0.68–1.08 | 0.5–0.6 | $\geq$0.5 | 0.6 | 0.7 | Variable in size | 0.62–0.86 |
| Head ventral chaetae along groove | 4+4 | 3+3 | ? | ? | ? | ? | 4–5+4–5 | 3+3 |
| Chaetae on ventral tube (anterior / distal /basal chaetae) | 1+1/7+7(8)/2+2 | 1+1/7+7/2+2 | ? | ? | ? | 1+1/7–8+7–8/2+2 | 1+1(2)/7–8+7–8/1–4+1–4 | 1+1/7+7/2+2 |
| Ratio of unguiculus/ unguis | 0.27–0.47 | 0.2 | 0.3–0.4 | 0.4 | 0.4 | Short, $\leq$0.25 after original drawing | Variable in size, mostly 0.5 | Vestigial, reduced to a minute, stumpy process |
| Chaetae on subcoxae 1 of legs I-III | 4–5, 4–5, 4–5 | 444 | 4/4–5/? | ? | ? | 4/4/? | ? | 4(3)44 |
| Location | France: Pointe-aux-oies | France: Pointe-aux-oies; Scotland: Dalmeny | France: Pointe-aux-oies | France: Pointe-aux-oies | France: Pointe-aux-oies | Spain: Pontevedra coast | Norwegian and Danish coast | Scotland: Dalmeny |

Non-type material from the type locality. France: Pas-de-Calais: Wimereux: Pointe-aux-Oies (1.361623°E, 50.463582°N), March 17, 2010, by hand and by washing of algae and sand, Sun Xin, Bedos A., Deharveng L. and Zon S. leg. (62-016, three males, three females, one juvenile on slides, including the skin of one male recovered after DNA extraction); same data (62-018, three males, one juvenile on slides, including the skins of one male and one juvenile recovered after DNA extraction). Ibid, August 5, 2010, by hand and by washing of algae and sand, Sun Xin leg. (62-044, three males, three females, two unsexed specimens, all on slides as skins of barcoded specimens recovered after DNA extraction); same data (62-045, five males, 12 females, two juveniles on slides, including the skins of one male, three females and one juvenile recovered after DNA extraction).

*Redescription:* Color: white. Length (without antennae): female 1.4–2.1 mm, male 1.4–1.65 mm. Body shape: cylindrical, slender, elongated, parallel-sided, with Abd. VI arched and anal spines 0.47–0.77 times as long as the inner edge of hind unguis (Figs. 2A, 2B and 3A). Granulation of body surface: regular, with more or less distinctly thinner granules on intersegment areas.

Pseudocelli is 32/1-233/3, 3-4, 3, 4-6, 3-4 dorsally, 11/000/0111(2)0 ventrally and 2/2/2 on subcoxae I–III (Figs. 3A, 3G and 4F). Parapseudocelli is absent. Pseudopores is 00/011/11110 dorsally, 00/111/000x0 ventrally (Figs. 3A, 3G and 4F).

S-chaetae not distinguishable from ordinary chaetae. S-microchaetae tiny and blunt, as 0/011/000000 dorsally (Fig. 3A).

The antennal basal area is not well delimited by granulation. The antennae are approximately 1.1 times as long as head. The length ratio of antennal segments I:II:III:IV is approximately 1.0:1.5:1.5:2.2. The antennal segment IV has subapical organite and basoexternal ms at approximately 1/3 length from the base (Fig. 3D). The Ant. III sensory organ is composed of five papillae, five guard chaetae, two small sensory rods and two smooth sense clubs (Fig. 3C). Ant. III has external ms just behind sensory organ (Fig. 3C). Ant. II has 13 chaetae. Ant. I has nine chaetae.

Postantennal organ is composed of 13–21 (16.0 ± 1.8 from 49 PAO) simple vesicles arranged in two rows along the axis of the organ (Fig. 3B). Dorsal cephalic chaeta $d_0$ is present (Fig. 3A). 3+3 chaetae appear between two inner posterior pso, and $p_1$ is anterior to others (Fig. 3A). The mandible has a strong molar plate and four apical teeth. The maxilla bears three teeth and six lamellae but is not examined in detail. The maxillary palp is simple with one basal chaeta and two sublobal hairs. The labral chaetae are 4/1, 4, 2. The labial papillae of AC type, papillae A–E are with one, four, zero, three and two guard chaetae, respectively (Fig. 3E). The labium has six proximal, four (E, F, G, and f) basomedial and six (a, b, c, d, e, e') basolateral chaetae. Postlabial chaetae are 4+4 along the ventral groove.

The ordinary chaetae were differentiated in macro- and meso-chaetae. Th. I has 7–9+7–9 dorsal chaetae with frequent asymmetries (Figs. 3A and 3F). Th. II–III has 4–5+4–5 dorsal chaetae and Abd. I–III has 3–4+3–4 dorsal chaetae along the axial line, usually symmetrically arranged but with differences between specimens. Abd. IV–V has dorsal chaetae asymmetrically arranged along the axis; Abd. VI with $m_0$ (Figs. 3A and 4F). Th. I–III has 1+1, 1+1 and 1+1 ventral chaetae, respectively, between the coxae.

Subcoxa 1 has 4–5, 4–5, 4–5 chaetae, and subcoxa 2 has 1, 4, 4 chaetae on legs I–III, respectively (Fig. 3A). Tibiotarsal chaetae are 18 (9, 8, 1), 18 (9, 8, 1) and 18 (9, 8, 1) chaetae on legs I–III, respectively (Figs. 4B–4D). The unguis is without teeth. The unguiculus is short, only 0.27–0.47 times as long as the inner edge the of unguis, with inner basal lamella (Figs. 4B–4D). The ventral tube has 1+1 anterior chaetae, 7+7 (rarely 7+8) distal chaetae and 2+2 basal chaetae (Fig. 4G). The furca was reduced to a finely granulated area, with four small chaetae in two rows posterior to the furcal rudiment (Figs. 3G and 4E).

The genital plate consists of 15–18 chaetae in female (Fig. 3G), 35–50 in male. The anal valves have numerous acuminate chaetae; each lateral valve with chaetae $a_0$ and $2a_1$; upper valve with chaetae $a_0$, $2b_1$, $2b_2$, $c_0$, $2c_1$, $2c_2$ (Fig. 4A).

*Habitats:* On the seashore, among *Fucus* and barnacles or under stones in the intertidal zone.

*Remarks*: The type material of *T. debilis* was in bad condition and only a few characters could be validated, i.e., the number of pso on Th. I tergum (2) and subcoxae I–III (2, 2, 2), the ratio of unguis/unguis (0.27–0.35), and the ratio of AS/unguis (0.55–0.57).

In the original description of the species by *Moniez (1890)*, the figure of the unguiculus corresponds to *T. debilis*, as redefined here (approximately 1/3 of claw length), as does the number of 23–28 vesicles in the PAO given in the text. The number of PAO vesicles in the Moniez' paratypes examined was not observable, but descriptions of the species by *Denis (1923)* based on eight syntypes of Moniez and by *Willem (1925)* and based on specimens from the type locality (Pointe-aux-Oies) state the number of vesicles as 20 and 17, respectively, which corresponds well with this redescription (Table 2). In the type locality, we found *T. debilis* was mixed with *T. thalassophila*, but in higher number. Therefore, it is possible that Moniez in 1890 included both species and described the unguiculus of a *T. debilis* and the PAO of a *T. thalassophila*.

Some characters of *T. debilis* are variable, especially the number of vesicles in the PAO (13–21), the number of dorsal pso (32/1-233/3, 3-4, 3, 4-6, 3-4), the number of pso on Abd. IV sternite (1 or 2), the length of unguiculus (0.27–0.47 times as long as the inner edge of unguis) and the number of chaetae on subcoxa 1 of legs (4–5). However, the length of unguiculus (short but clearly longer than that of *T. thalassophila*) and the presence of pseudocelli on the abdominal sterna allow separation of *T. debilis* from *T. thalassophila*. *Fjellberg (1998)* emphasized the former character in his work but apparently did not consider it as having a taxonomic value.

THALASSAPHORURA THALASSOPHILA (*BAGNALL, 1937*)

(Figs. 2, 5 and 6; Table 2)

*Onychiurus thalassophilus* Bagnall (1937): 146)
*Onychiurus imminutus* Bagnall, 1937: 146 after *Salmon (1959*: 149, *syn. dub.)*
*Thalassaphorura thalassophila* in *Bagnall (1949*: 504)
*Onychiurus thalassophilus* in *Stach (1954*: 44)

*Spelaphorura thalassophilus* (sic) in *Salmon (1959*: 149*)*

*Protaphorura debilis* in *Jordana et al. (1997*: 571*)*

*Thalassaphorura thalassophila* in *Pomorski (1998*: 135*)*

*Thalassaphorura debilis* in *Fjellberg (1998*: 109*)* (synonymy not accepted here)

*Material examined:* Type material (examined). Three female syntypes of the Bagnall type series. Great Britain, Scotland: Dalmeny Estate shore, well below the high-water mark, 12. V.35 (deposited in The Natural History Museum, London).

Non-type material examined. Great Britain, Scotland: Dalmeny Estate shore (3.310991°E, 55.983110°N), April 05, 2016, by hand and by washing of algae and sand, Sun Xin, Bedos A. and Deharveng L. (GB-011, four males, 11 females and one unsexed specimen on slides, including the skins of one male, two females and one unsexed specimen recovered after DNA extraction). France: Pas-de-Calais: Wimereux: Pointe-aux-Oies (1.361623°E, 50.463582°N), March 17, 2010, by hand and by washing of algae and sand, Sun Xin, Bedos A., Deharveng L. and Zon S. leg. (62-016, two males and one female on slides, including the skin of one female recovered after DNA extraction); same data (62-017, one male, one female and one unsexed specimen on slides, including the skins of one male and one unsexed specimen recovered after DNA extraction). Ibid, August 5, 2010, by hand and by washing of algae and sand, Sun Xin leg. (62-044, the skin on slide of one female recovered after DNA extraction); same data (62-045, three males, three females and three juveniles on slides, including the skin of one juvenile recovered after DNA extraction).

*Redescription:* Color: white. Length (without antennae): female 1.32–1.93 mm; male 1.20–1.66 mm. Body shape: cylindrical, slender, elongated, parallel-sided, with Abd. VI arched and anal spines 0.68–1.08 times as long as the inner edge of hind unguis (Figs. 2C, 2D and 5A). Granulation of body surface: regular, with more or less distinctly thinner granules on intersegment areas.

Pseudocelli as 32/133/33343 dorsally, 11/000/00000 ventrally and 1/1/1 on subcoxae I–III (Figs. 5A, 5F, 6A and 6F). Parapseudocelli absent. Pseudopores as 00/011/11110 dorsally, 00/111/000x0 ventrally (Figs. 5A, 5F, 6A and 6F).

The S-chaetae is not distinguishable from ordinary chaetae. The S-microchaetae is tiny and blunt, as 0/011/000000 dorsally (Fig. 5F).

The antennal basal area is not well delimited by granulation. The antennae are as long as the head. The length ratio of antennal segments I:II:III:IV is approximately 1.0:1.2:1.2:1.8. The antennal segment IV with subapical organite and basoexternal ms is at approximately 1/3 length from the base. The Ant. III sensory organ is composed of five papillae, five guard chaetae, two small rods and two smooth clubs (Fig. 5D). Antennal segment III has external ms just behind sensory organ (Fig. 5D). Ant. II has 13 chaetae. Ant. I has nine chaetae.

The PAO is composed of 16–23 ($19.9 \pm 1.7$ from 48 PAO) simple vesicles arranged in two rows along the axis of the organ (Fig. 5C). The dorsal cephalic chaeta $d_0$ is present (Fig. 5A). 3+3 chaetae appear between two inner posterior pso, while $p_1$ is anterior to others (Fig. 5A). The mandible has a strong molar plate and four apical teeth. The maxilla bears three teeth

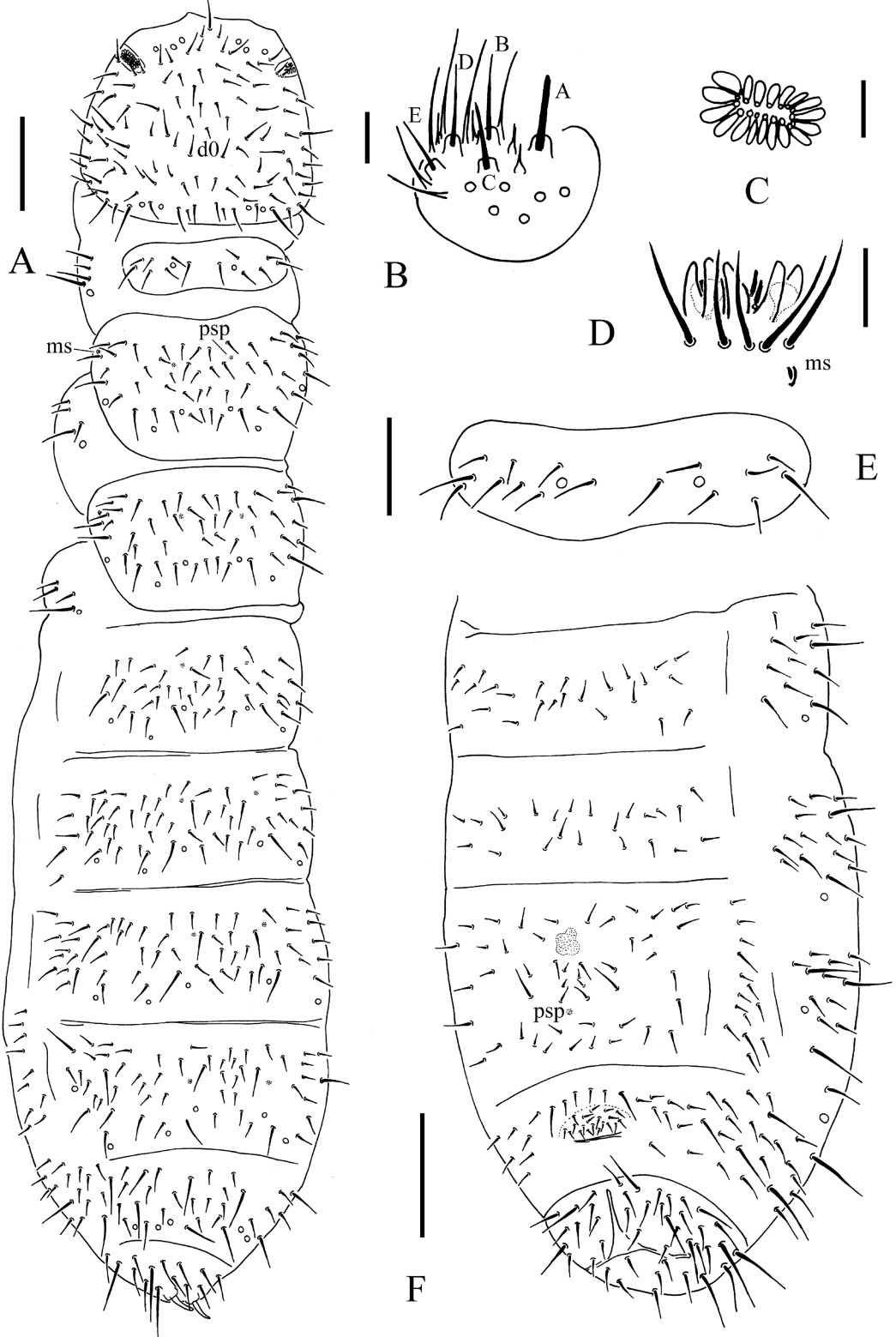

**Figure 5** ***T. thalassophila.*** (A) Habitus, pseudopores and dorsal chaetotaxy of head and body. (B) Labium. (C) Postantennal organ. (D) Ant. III sensory organ. (E) Th. I tergum. (F) Abd. II–VI sterna. Scales: 0.1 mm (A, F), 0.05 mm (E), 0.01 mm (B, C, D).

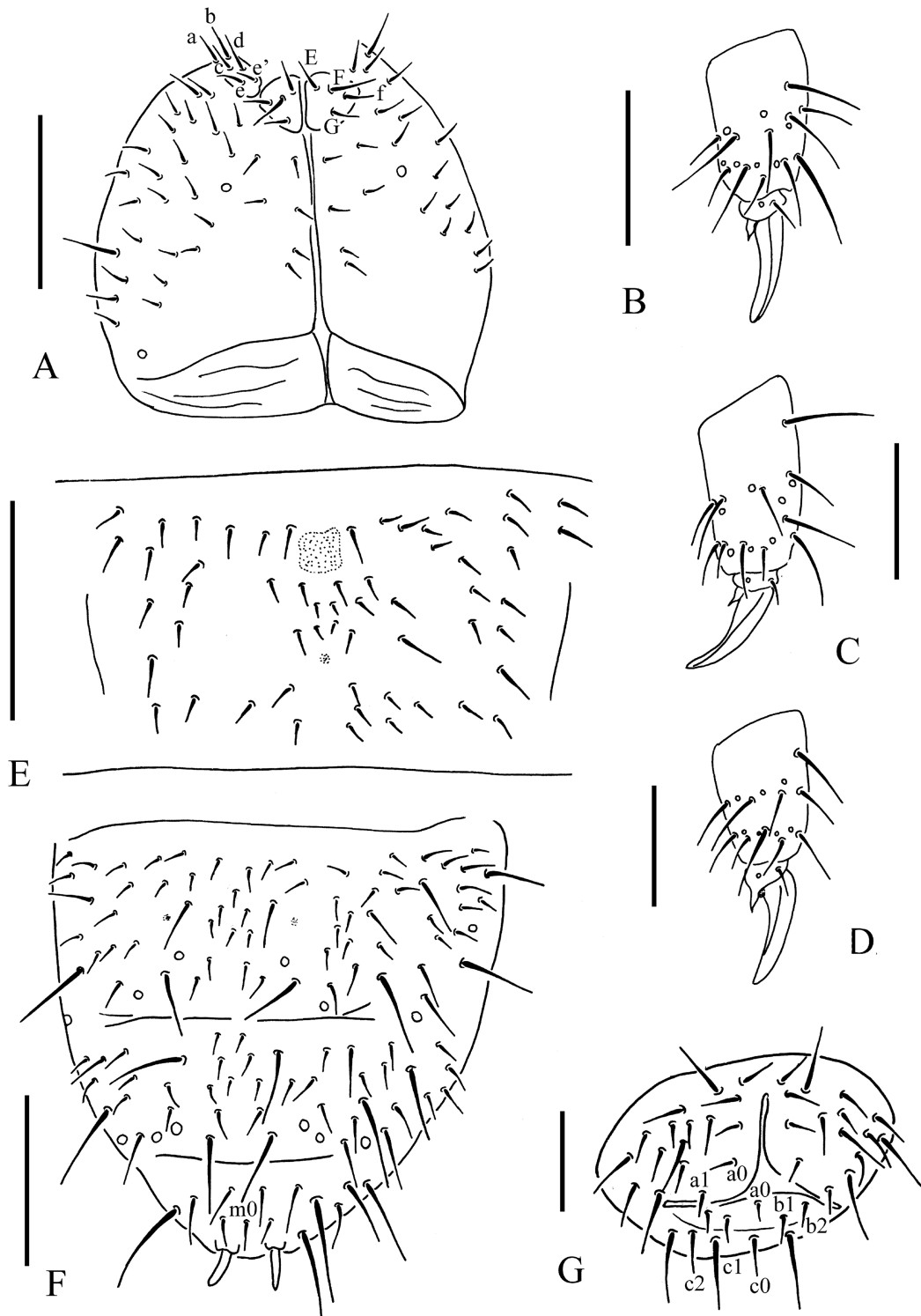

**Figure 6 _T. thalassophila._** (A) Ventral side of head. (B) Tibiotarsal chaetotaxy and claw of leg I. (C) Tibiotarsal chaetotaxy and claw of leg III. (D) Tibiotarsal chaetotaxy and claw of leg III (type material). (E) Abd. IV sternum. (F) Abd. IV–VI terga. (G) Anal valves. Scales: 0.1 mm (A, E, F), 0.05 mm (B, C, D, G).

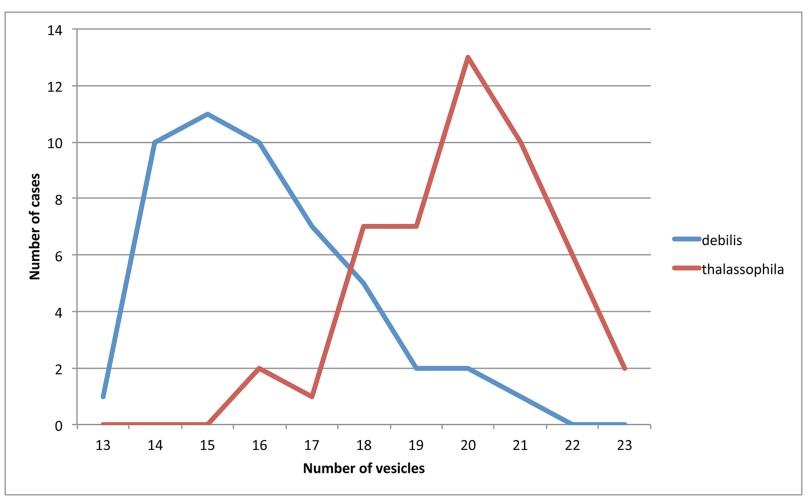

**Figure 7 Number of observations (ordinates) for different numbers of PAO vesicles (abscissa) in *T. debilis* and *T. thalassophila*.**

and six lamellae but was not examined in detail. The maxillary palp is simple with one basal chaeta and two sublobal hairs. The labral chaetae are 4/1, 4, 2. The labial papillae are of AC type, papillae A–E with one, four, zero, three and two guard chaetae, respectively (Fig. 5B). The labium has six proximal, four (E, F, G, and F) basomedial and six (a, b, c, d, e, e') basolateral chaetae (Fig. 6A). The postlabial chaetae are 4+4 along the ventral groove.

Ordinary chaetae were differentiated in macro- and meso-chaetae. Th. I has 6–7+6–7 dorsal chaetae (frequent asymmetries) (Figs. 5A and 5E). Th. II–Abd. III has 3–4+3–4 dorsal chaetae along the axial line, usually symmetrically arranged but with differences between specimens. Abd. IV–V has dorsal chaetae asymmetrically arranged along the axis. Abd. VI has $m_0$ and sometimes $a_0$ present (Figs. 5A and 6F). Th. I–III has 1+1, 1+1 and 1+1 ventral chaetae, respectively, between coxae.

Subcoxa 1 has 4, 4, 4 chaetae, and subcoxa 2 has 1, 4, 4 chaetae on legs I–III, respectively (Fig. 5A). Tibiotarsal chaetae has 18 (9, 8, 1), 18 (9, 8, 1) and 18 (9, 8, 1) on legs I–III, respectively (Figs. 6B–6D). The unguis is without teeth. The unguiculus very short, reduced to a minute, stumpy process and is 0.2 times as long as the inner edge of the unguis, with inner basal lamella (Figs. 6B–6D). The ventral tube has 1+1 anterior chaetae, 7+7 distal chaetae, and 2+2 basal chaetae. The furca is reduced to a finely granulated area, with four small chaetae in two rows posterior to the furcal rudiment (Figs. 5F and 6E).

The genital plate consists of 18–21 chaetae in females (Fig. 5F), and 40–42 in males. The anal valves have numerous acuminate chaetae; each lateral valve has chaetae $a_0$ and $2a_1$; the upper valve has chaetae $a_0$, $2b_1$, $2b_2$, $c_0$, $2c_1$, $2c_2$ (Fig. 6G).

*Habitats*: Similar to *T. debilis*, on the seashore, among *Fucus* and barnacles or under stones in the intertidal zone.

*Remarks: T. thalassophila* is very similar to *T. debilis* by its habitus, non-differentiated dorsal S-chaetae, and short unguiculus. However, it can be easily distinguished by several characters (Table 2): it has shorter unguiculus, reduced to a minute and stumpy process; the papillae of AIIIO are longer and slender; there are usually more vesicles in

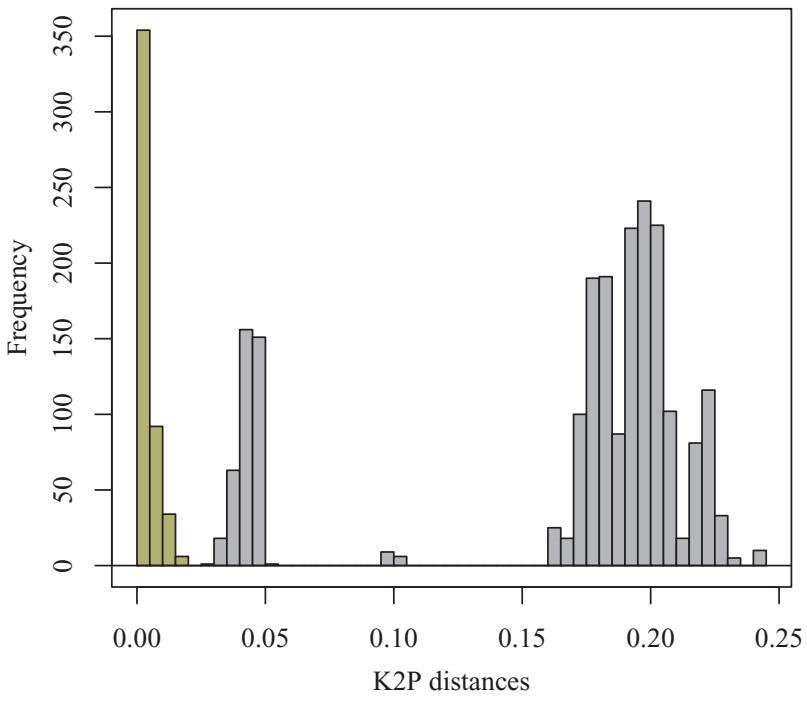

**Figure 8 Frequency histogram of K2P pairwise distances.** Columns of the intra-specific divergences are greenish-yellow colored.

PAO (Fig. 7) there are no pso on the abdominal sterna; there are fewer chaetae on the subcoxae; and the AS is usually longer. We did not find significant intra-specific variations in the pso formula, and the size of the unguiculus among the studied specimens of *T. thalassophila* is contrary to those of *T. debilis*. *P. debilis* as redescribed by *Jordana et al. (1997*: 571*)* on Spanish material is probably *T. thalassophila* according to the diagnostic characters, except for the number of PAO vesicles, which could correspond to another species.

Overall, *T. debilis* and *T. thalassophila* represent two species that are closely related but morphologically clearly distinct based on standards of modern Onychiuridae taxonomy (*Pomorski, 1998*). Therefore, the two taxa are not synonymous as proposed by *Fjellberg (1998*: 109*)* (the author described differences in unguiculus size between the two species, but did not consider them to be sufficient for separating the species).

### Barcode characterization of the two species

In total, 16 (62% of barcoded specimens of *T. debilis*) and nine (60% of barcoded specimens of *T. thalassophila*) individuals were examined for morphological diagnostic characters after DNA extraction (Table S1). The remaining specimens were damaged during DNA extraction and were therefore morphologically uninformative.

A small barcoding gap was observed at K2P distances of approximately 0.02 (Fig. 8). The two species *T. debilis* and *T. thalassophila* are clearly characterized by their barcode (Fig. 9), with a small inter-specific divergence of 4.3% and intra-specific divergence of 0.49% (0–1.9%) in *T. debilis* and 0.16% (0–0.3%) in *T. thalassophila* (Fig. S1; Tables 3 and 4).

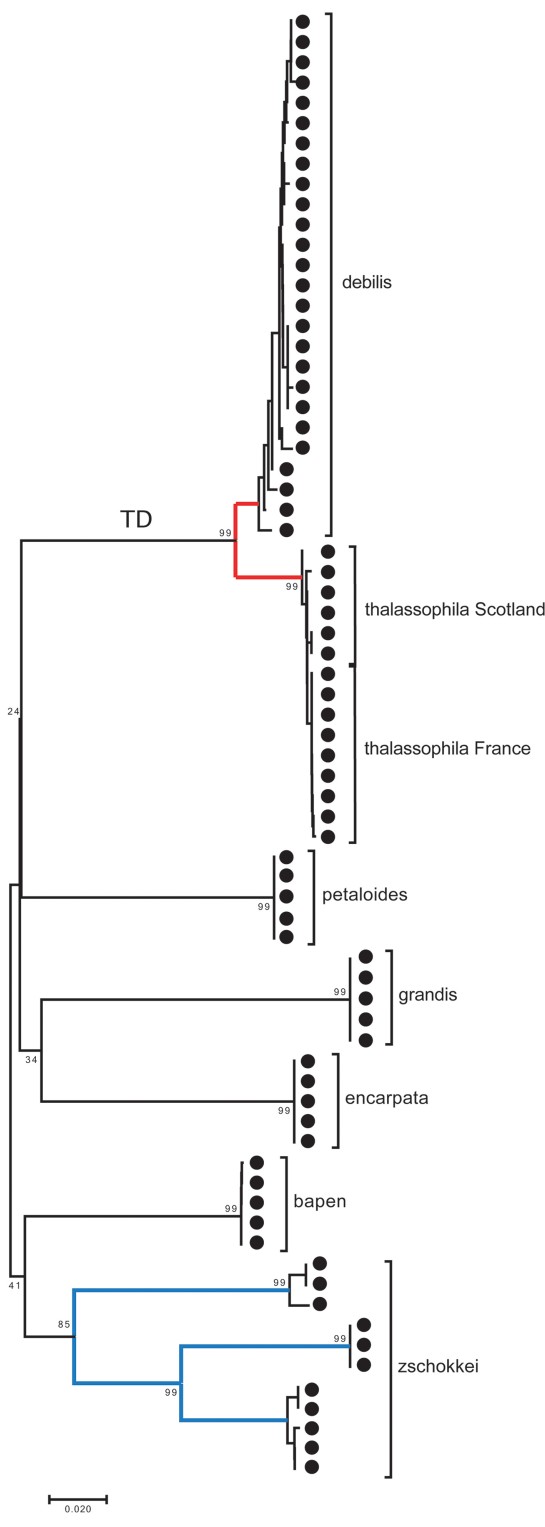

**Figure 9 Neighbor-joining tree (K2P) based on COI for the seven *Thalassaphorura* species, including three clusters of *T. zschokkei*.** The numbers at MOTU nodes are bootstrap values above 80% (1,000 replicates). TD: the branch of *Thalassaphorura debilis* and *T. thalassophila*.

**Table 3 Intraspecific and intra-MOTUs divergence within the genus *Thalassaphorura*.**

| Species | Intraspecific divergence |
|---|---|
| *Thalassaphorura grandis* | 0 |
| *Thalassaphorura debilis* | 0.004900574 |
| *Thalassaphorura thalassophila* | 0.001593864 |
| *Thalassaphorura petaloides* | 0 |
| *Thalassaphorura bapen* | 0 |
| *Thalassaphorura zschokkei* | 0.102789204 |
| *Thalassaphorura encarpata* | 0 |

**Table 4 Molecular divergence (COI) between *Thalassaphorura* species (A), between populations of the *T. debilis–T. thalassophila* group (B), and between three populations of *T. zschokkei* (C).**

**A**

| Species | *bapen* | *debilis* | *encarpata* | *grandis* | *petaloides* | *thalassophila* |
|---|---|---|---|---|---|---|
| *debilis* | 0.179 | | | | | |
| *encarpata* | 0.163 | 0.191 | | | | |
| *grandis* | 0.220 | 0.204 | 0.196 | | | |
| *petaloides* | 0.174 | 0.179 | 0.181 | 0.206 | | |
| *thalassophila* | 0.181 | 0.043 | 0.194 | 0.223 | 0.191 | |
| *zschokkei* | 0.176 | 0.205 | 0.191 | 0.226 | 0.199 | 0.205 |

**B**

| Species | *debilis* | *thalassophila_FR* |
|---|---|---|
| *thalassophila_FR* | 0.044 | |
| *thalassophila_SC* | 0.042 | 0.003 |

**C**

| Species | *zschokkei_1* | *zschokkei_2* |
|---|---|---|
| *zschokkei_2* | 0.167 | |
| *zschokkei_3* | 0.100 | 0.170 |

**Note:**
FR, France; SC, Scotland.

The two populations of *T. thalassophila* (France and Scotland) show a very low divergence (0.03%). The non-intertidal species of *Thalassaphorura* exhibited much higher values of inter-specific divergence (from 16.3% between *T. bapen* and *T. encarpata* to 22.6% between *T. grandis* and *T. zschokkei*), and very low intra-specific divergence, except in *T. zschokkei* (10.28%), which is split in well-separated MOTUs that are morphologically indistinguishable (Tables 3 and 4). Divergence time estimation indicated that the speciation event of the two species *T. debilis* and *T. thalassophila* occurred at 1.66 (0.47–3.14) Mya (Fig. S2).

## DISCUSSION

In the present study, we used specimens from the type localities of *T. debilis* and *T. thalassophila*, as the state and age of the type material on slides that precluded

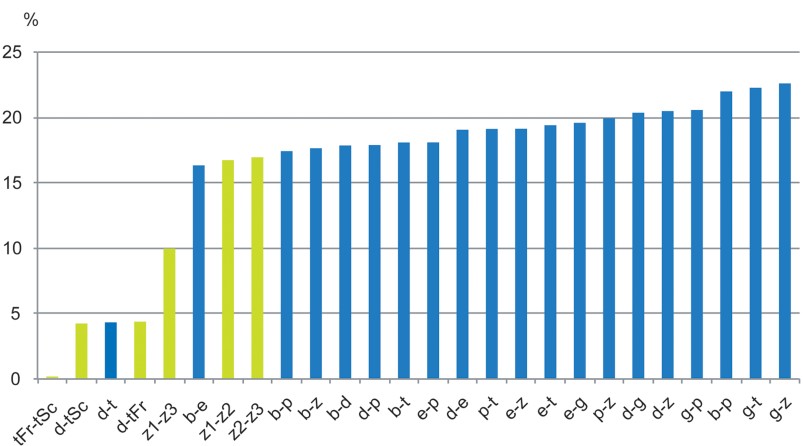

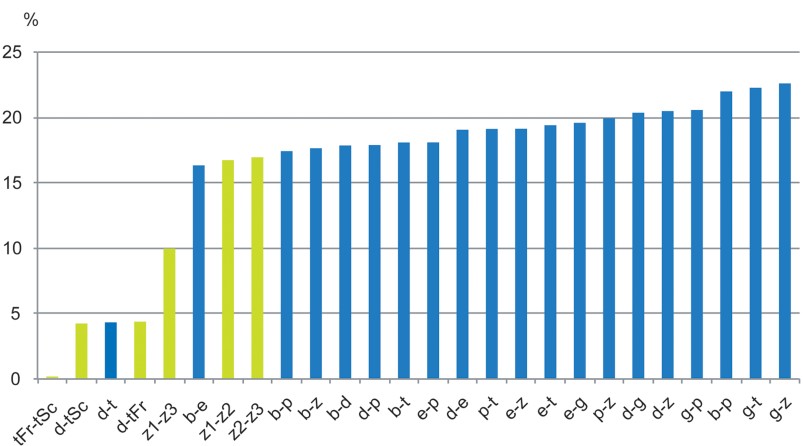

**Figure 10 Histograms of COI divergence in % between species, MOTUs of *Thalassaphorura*.** In green, between species and populations of the intertidal species *debilis—thalassophila*, and between three MOTUs of the edaphic species *T. zschokkei*; in blue, between edaphic species of the genus, and between them and the two intertidal species. Study sites: b, *bapen*; d, *debilis*; e, *encarpata*; g, *grandis*; p, *petaloides*, t, *thalassophila*; z, *zschokkei* (with three MOTUs: −1, −2, −3); Fr, France; Sc, Scotland.

extraction of reliable genetic material. The combined genetic and geographic pattern of the three analyzed populations (*T. debilis*, *T. thalassophila* France and *T. thalassophila* Scotland) can be summarized as follows (Figs. 9 and 10; Table 3): (i) moderate but clear molecular divergence between *T. debilis* (France) and *T. thalassophila* (France and Scotland); (ii) very low molecular divergence between *T. thalassophila* from France and *T. thalassophila* from Scotland in spite of the geographic distance between them; and (iii) co-occurrence in syntopy of *T. debilis* and *T. thalassophila* in France.

Morphology and genetic data were congruent in support for the species status of both taxa. However, the low level of genetic divergence between *T. debilis* and *T. thalassophila* was unusual when compared to genetic differences usually observed between congeneric species of Collembola (Tables 3 and 4). Low genetic divergence associated with clear morphological differences is reported here for the first time in Collembola (*Porco et al., 2012a, 2013*). In *Deutonura zana Deharveng, Zoughailech, Hamra-Kroua & Porco (2015)*, for instance, two populations geographically separated and genetically divergent at 3.7% did not reveal any morphological difference despite a thorough examination (*Deharveng et al., 2015*).

For other species within the genus *Thalassaphorura*, the interspecific divergences we measured were in line with the high values observed for other Collembola, ranging from 16.4% to 22.6% between all couples of the five non-marine species, as well as between these species and each intertidal species (Table 4). The low divergence between *T. debilis* and *T. thalassophila* was more similar to that among many winged arthropods and lower than that among three populations of closely related, morphologically indistinguishable non-marine species (Figs. 9 and 10; Table 4). This unusual pattern may be the result of our failure to detect discriminant morphological characters between populations of this last species. It also reflects different paces of morphological and molecular
diversification among the *Thalassaphorura* species, which would potentially impact our understanding of intra- versus inter-specific variations among Collembola. Biologists using approaches for MOTU delimitations based on a barcode gap approach, e.g., ABGD (*Puillandre et al., 2012*), or on the use of a threshold derived from empirical data should be aware of such cases that may cause underestimation of actual diversity, as some species get overlooked.

The frequency of occurrence of the observed patterns is unknown and its origin obscure. It is probably not linked to phylogeny, as other *Thalassaphorura* species (Table 4) have divergences similar to other Collembolan genera. Furthermore, the estimated divergence time (0.47–3.14 Mya) between the two species is small compared to other species, suggesting that *T. debilis* and *T. thalassophila* could be two young sister species. The calibration method applied here is not optimal, as it is based, in the absence of biogeographically informative pattern, on Tenebrionidae beetles which probably have a much longer life cycle than *Thalassaphorura*. However, the *T. zschokkei* populations as well as other species of the genus analyzed here would have diverged much earlier. Therefore, the inference of a lower evolutionary pace of the *T. debilis–T. thalassophila* lineage cannot be ruled out. Because of these uncertainties, as well as the sympatric occurrence of the two species, the time of divergence for the two species cannot reliably be inferred.

High divergence in COI sequences between geographically distant MOTUs of the same morphological species is frequent in Collembola (*Porco et al., 2013*), especially among non-widespread species. This is illustrated in the dataset analyzed by *Porco et al. (2012a)*, where populations of several species drawn from various Collembolan families were represented by MOTUs, which diverged from conspecific MOTUs by 11.33–21.47% (with less than 2% intra-population divergence), matching, in most cases, the levels of divergence observed between congeneric species of Collembola. This may indicate the presence of yet unrecognized species, especially where the different MOTUs were found in sympatry. However, in several cases, such as for *Bilobella aurantiaca* (*Caroli, 1912*), thorough morphological analysis did not reveal morphological differences between conspecific MOTUs. We observed similarly high levels of divergence without morphological differentiation between three MOTUs of the non-marine species *T. zschokkei* (Fig. 10; Table 4), which were from populations 40–85 km apart and spread across the Southern Alps. Conversely, the two populations of *T. thalassophila* studied were 660 km apart (Fig. 1), but did not show genetic divergence at COI, which is similar to divergences often observed among widely distributed species that are suspected to be dispersed by humans (*Porco et al., 2013*). The common assumption is that marine currents might be a powerful dispersal agent for flightless littoral arthropods (*Hawes et al., 2008*; *Witteveen & Joosse, 1988*), maintaining gene flow and explaining the very low genetic differentiation observed between populations. However, the link between wide distribution with efficient dispersal by ocean currents and low genetic divergence among populations is yet to be clearly documented for intertidal species.

The co-occurrence of two closely related species in the same microhabitat without apparent niche or trait differentiation is unusual. The two species are similar, and

their minor morphological differences are probably not ecologically significant. Co-occurrences of genetically closely related and morphologically highly similar species are unknown among Collembola. When co-occurrences of morphological similar species have been reported, the taxonomic status of the species was uncertain, their microhabitat was slightly different (*Rusek, 2007*), or their distribution only overlapped in a narrow strip in a contact zone between parapatric forms (*Deharveng, Bedos & Gisclard, 1998*). Therefore, the co-existence of the morphologically similar *T. debilis* and *T. thalassophila* in the same habitats should be further investigated.

The only evident biological feature that strongly separates our two species from non-marine *Thalassaphorura* is their peculiar intertidal ecology, as stressed above. Whether the *debilis/thalassophila* case is representative of genetic patterns associated with this environment will have to be investigated in other Collembola. However, aside from the intertidal species group of *Anurida maritima,* very few genera or species groups are known to involve marine and non-marine species and to encompass closely related intertidal forms.

## ACKNOWLEDGEMENTS

We thank Serge Zon from the Cocody University (Abidjan, Ivory Coast) for his help in field sampling; Wanda Maria Weiner from the Polish Academy of Sciences (Krakow) for helpful advice on the species taxonomy; Paul Brown from The Natural History Museum, London for the loan of the type material of *T. thalassophila*; David Porco from the Musée National d'Histoire Naturelle, Luxembourg; Marianne Elias and Rodolphe Rougerie from the Muséum National d'Histoire Naturelle, Paris; Feng Zhang from Nanjing Agricultural University for useful comments during the preparation of the manuscript; and Gunnar Keppel from University of South Australia, Andrew Davis from University of Goettingen for the language modification.

### Funding

The present study was supported by funding from the National Natural Science Foundation of China (Grant No. 41571052, 41430857), Science and Technology Development Plan Project of Jilin Province (20160520051JH), the Postdoctoral Science Foundation of China (Grant No. 2015M570281), the Excellent Young Scholars of Northeast Institute of Geography and Agroecology, Chinese Academy of Sciences (Grant No. DLSYQ13003), the Alexander von Humboldt Foundation, Youth Innovation Promotion Association, CAS, the European project EDIT and the Parc National du Mercantour. There was no additional external funding received for this study. The funders had no role in study design, data collection and analysis, decision to publish, or preparation of the manuscript.

### Grant Disclosures

The following grant information was disclosed by the authors:
National Natural Science Foundation of China: 41571052, 41430857.

Science and Technology Development Plan Project of Jilin Province: 20160520051JH.
Postdoctoral Science Foundation of China: 2015M570281.
Excellent Young Scholars of Northeast Institute of Geography and Agroecology, Chinese Academy of Sciences: DLSYQ13003.
Alexander von Humboldt Foundation.
Youth Innovation Promotion Association.
CAS.
European project EDIT.
Parc National du Mercantour.

## Competing Interests

The authors declare that they have no competing interests.

## Author Contributions

- Xin Sun conceived and designed the experiments, performed the experiments, analyzed the data, contributed reagents/materials/analysis tools, prepared figures and/or tables, authored or reviewed drafts of the paper, approved the final draft.
- Anne Bedos performed the experiments, analyzed the data, authored or reviewed drafts of the paper, approved the final draft.
- Louis Deharveng conceived and designed the experiments, performed the experiments, analyzed the data, contributed reagents/materials/analysis tools, prepared figures and/or tables, authored or reviewed drafts of the paper, approved the final draft.

## DNA Deposition

The following information was supplied regarding the deposition of DNA sequences:

The sequences described here are accessible via BOLD sequence numbers which have been provided in the Supplemental Table.

## Data Deposition

The raw data for the main diagnostic characters of the two species are provided as a Supplemental File.

## Supplemental Information

Supplemental information for this article can be found online at http://dx.doi.org/10.7717/peerj.5021#supplemental-information.

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
