# Peer review of "Unusually low genetic divergence at COI barcode locus between two species of intertidal Thalassaphorura (Collembola: Onychiuridae)"

_PeerJ, doi:10.7717/peerj.5021_

## Round 0.1 · original submission · Major Revisions

I have heard back from two reviewers. Both were generally favourable towards your manuscript, but both do wish to see more work on the fine-tuning of your text, and I find their reviews to be constructive and helpful in nature. In general, your analyses seem to be sound, and both authors ask for more text and information (e.g. back story) and cleaning up of your English. It may serve you well to edit the paper to the best of your ability, and then have a colleague or editing service go over this once before any resubmission. I look forward to seeing a revised version of your work.

Reviewer 1 ·

Basic reporting

a. There is some room for improvement concerning style, prose, grammar, and structure of the article.
i. In general, the methods sections seems out of order. For example, the discussion of preparing specimen vouchers that were recovered post-DNA extraction before the sections on molecular analysis is confusing. Consider presenting the methods in a more logical/sequential order to improve flow of the section.
ii. The methods also includes sub headers which are redundant, and provide slightly contrasting information. The “Molecular analysis sub section suggests that molecular analysis follows Porco et al. (which only describes recovery of vouchered specimens following DNA extraction) while the DNA Extraction sub section describes in full (and correct) detail the actual methods of molecular analysis. I suggest collapsing these two sections together and revising for clarity and cohesiveness.
iii. The Data analysis sub section of the methods also requires some editing, as it lacks cohesion and is difficult for the reader to follow.
iv. The description of DNA barcoding in the introduction seems a bit pedestrian. More discussion on the gene region that is commonly used, and the strengths and weaknesses of the method would be helpful to the reader.
v. Overall, some sections are very well written while others use colloquial language and are unpolished. Below are several suggestions for correcting grammar or improving style, but is by no means an exhaustive list.


Line1: suggested edit to title “Unusually low” rather than “Unusual low”
Line 21: suggested edit: “combined morphology and DNA barcode data”
Lines 25-29: the use of the word “weaker” suggests, strength of signal rather than size of divergence. Suggested edit: “much smaller than previously reported in Collembola”. These two sentences could be further strengthened if edited to be more concise.
Line 44: suggested edit: “large numbers” or “in very high abundance”.
Line 78: suggested edit: “firmly setting” rather than “setting fimly”
Line84: “DNA barcoding” rather than “DNA barcode”
Line 86: “is at pain” this seems rather odd terminology. I suggest rephrasing
Line 113: “strictly restricted” is redundant
Line 140: “Dewaard” should be spelled “deWaard”
Line 400: “less than 2% of intra-population divergence” –remove the “of” from the sentence.
Line 422: I think you mean “strip” rather than “stripe”
Line 427: use “genetic” rather than “genetics”
Line 428-430: I don’t really understand what you mean here

Experimental design

a. Overall I think the experimental design is valid and well expressed. There are a few questions raised which could be clarified.
i. Line 127: how many specimens were sent for DNA barcode analysis in total? Did you send all specimens for DNA barcode analysis, or did you choose some for barcode and some for morphology separately?
ii. I’m not sure if this is a community standard, but there doesn’t appear to be any description of the re-examination of type material in the methods section. Is that normal for manuscripts which revise species descriptions?

Validity of the findings

a. I would like the authors to consider biological reasons why the two species in question have smaller interspecific divergences than were expected. It is important to consider the age of the species (perhaps these are two young sister-species?), the rate of accumulation of mutation (could these taxa be evolving more slowly at COI for some reason?). I think it’s important to place the sequence divergence results more clearly into biological context.
b. Use of the work “weak” in reference to your barcode results is disheartening. The results clearly discriminate both species, with very low intraspecific divergence detected. Using the word weak suggests to the reader that the relationship is weak, but it is rather clear in your results. I would highlight this, and replace the work weak with something else. Perhaps the word small. You should place more emphasis on the fact that a barcode gap was clearly detected.
c. Figure 9 – this figure is not really providing a whole lot to the reader. It might be more useful to provide the typical barcode gap graph which depicts the maximum intraspecific divergence on the x axis and the minimum intraspecific divergence on the y axis. This will allow you to highlight the taxa which have, smaller/larger barcode gaps in a more intuitive way.

Reviewer 2 ·

Basic reporting

The authors did well in the language and structure of the text. However, I found that the background might not be sufficiently introduced. I suggest adding a brief introduction to the taxonomic difficulty of the objective group.

Experimental design

no comment

Validity of the findings

The main aim of the study is to clarify that T. thalassophila and T. debilis are independent species. The results showed that neither morphological nor molecular divergence is pronounced between the two forms, therefore only the combination of them can perhaps justify the two species. However, in the text the two parts are treated separately, which weakened the validity of evidences. I suggest an expanding of the beginning of the 4th paragraph in the discussion section and raise it up to the beginning of discussion, because it is basic and most important that the morphological and molecular evidences are congruent.

Additional comments

Some comments are made in the text. In general, some parts of the text need to be rearranged to achieve a logical coherence.

Annotated reviews are not available for download in order to protect the identity of reviewers who chose to remain anonymous.

---

## Round 0.2 · Minor Revisions

The paper is much improved, and almost ready for acceptance. Still, one reviewer has noted some areas where improvement of the English is needed, and I agree with their comments. As the amount of edits is more than we would like from a proof, I am returning this to you with minor revisions.

Reviewer 1 ·

Basic reporting

a. There is some room for improvement concerning prose and grammar
i. Overall, some sections are very well written while others use colloquial language and are unpolished. Below are several suggestions for correcting grammar and improving clarity

Line 47: “in intertidal environment” suggest “in the intertidal environment”
Line 97: “are regarded powerful tools” suggest “are regarded as powerful tools”
Line 103: “or Lepidoptera” suggest “and Lepidoptera”
Line 104-105: “High percentage divergence in DNA sequences provides…” Suggest “Large divergences (>5%) in DNA barcode sequences provides..”
Line 107 – 110 “Insects usually have lower divergence than non-winged arthropods, and percentage divergence is 11.5% in Hymenoptera and 9.3% in Diptera (Hebert, Ratnasingham & deWaard, 2003), 13.9% in Ephemeroptera of North America (Webb et al., 2012) and 7–8% in holarctic Lepidoptera (Hebert 110 & Landry, 2010, Hausmann et al., 2011).”
Suggest “Insects usually have lower interspecific divergences than non-winged arthropods. For example, average DNA barcode distances between congeneric species range from 7–8% in holarctic Lepidoptera (Hebert 110 & Landry, 2010, Hausmann et al., 2011) and 9.3% in Diptera (Hebert, Ratnasingham & deWaard, 2003), to 11.5% in Hymenoptera and 13.9% in North American Ephemeroptera (Webb et al., 2012).
Line 115 – 118: “Furthermore, recent molecular studies amongst Collembolan lineages have revealed divergences almost as deep as amongst congeneric morphological species (Cicconardi, Fanciulli & Emerson, 2013; Emerson et al., 2011; Frati et al., 2000; Katz, Giordano & Soto-Adames, 2015; Porco et al., 118 2012b; Soto-Adames, 2002).” It is not clear here what you mean by Collembolan lineages – do you mean divergences within species?
Line 392: “clear distinct…” suggest “clearly distinct…”
Line 393: “not synonymys roposed by Fjellberg” suggest “not synonymous as proposed by…”
Line 400: “The remaining specimens were damaged during the extraction” suggest adding “… damaged during DNA extraction and therefore morphologically uninformative”
Line 401: “A small barcoding gap was observed at K2P distances of around 0.02 (Fig. 8).” I understand the visual you are trying to express here, but the bars should be different colors to show how the intra-specific divergences are distributed in comparison with the inter-specific divergences, as illustrated in Figure 2 of Meyer and Paulay 2005 - http://journals.plos.org/plosbiology/article?id=10.1371/journal.pbio.0030422). In fact – this particular type of figure is far less in formative than if you showed a scatter plot of the maximum intraspecific divergence on the x axis and minimum distance to the nearest congeneric species on the y-axis of each species, similar to figure 2 in Blagoev et al 2016 (https://onlinelibrary.wiley.com/doi/full/10.1111/1755-0998.12444).
Line 444: “have divergence level” suggest “have divergences similar…”
Line 458: “which diverged from conspecific MOTUs from 11.33 to 21.47%” suggest “..MOTUs by 11.33 to 21.47%”
Line 463: “We observed similar high levels” suggest “… similarly high levels”
Line 467: “but did not diverge genetically for COI sequences” suggest “but did not show genetic divergence at COI”

Experimental design

Overall I think the experimental design is valid and well expressed. No comments here.

Validity of the findings

The research is integrated into the literature well and attempts to explore biological reasons for the unexpectedly low divergence between these two inertial species.

Additional comments

Good job addressing the previous concerns, This paper is much closer to a final form that could be published.

Reviewer 2 ·

Basic reporting

Additional background has been added in the introduction, no more comment

Experimental design

no comment

Validity of the findings

Changes have been made, no more comment

Additional comments

The authors made some changes in the arrangement and expression of the text except some part of the description. They explained the reason and I find there is no problem that they insist on their own style. No more comment.

---

## Round 0.3 · Minor Revisions

There are still yet a few more areas that need English improving, more than the specific comments previous reviewer has mentioned. I do thank you for correcting the previous edits, but at the same time, a check of the total paper would have been better. I apologize, I should have explicitly asked for this. However, the final English of the paper is the authors' responsibility, as we do not have editors for such things at PeerJ. If there are a very few edits I will usually accept a paper, but think there are slightly more than "a very few" and thus return this to you for one more round of edits.

I suggest a professional editing service or a college to help proofread your paper one last time.

I have checked the Abstract and Introduction for you, to give you an idea of level and type of edits that remain:

line 30 "that either of them deserves a specific status" is odd, please rephrase.
line 31: I am not sure if this makes sense: "However, their morphological distinctiveness was supported by a molecular divergence much smaller than previously reported at interspecific level amongst Collembola. " the small molecular divergence should NOT support the morphological distinctiveness in my opinion.
line 52: comma after "Bellinger et al. (2015)" needed
line 65: you confuse words, "confusing" should be "confused"
line 74: "on its side" is odd phrasing, can be deleted.
line 87: You say "More than sixty species have been assigned to the genus until now " but earlier you say "57".
line 92: Add "the" before "dominant".
line 125: "the two long doubtful species " is awkward, please reword.

Please note I have not checked the remainder of the paper, and trust you can find a service or colleague to thoroughly go over this one more time.

---

## Round 0.4 · accepted · Accept

The language revision is generally well done. Note that there are still some small edits required - I have added these to an annotated MS Word file. Please ensure all corrections are done at the proof stage or earlier.

I look forward to seeing your work in its published form,

#